# Fourth-Order Difference Scheme and a Matrix Transform Approach for Solving Fractional PDEs

Zahrah I. Salman [1], Majid Tavassoli Kajani [1,*], Mohammed Sahib Mechee [2] and Masoud Allame [1]

[1] Department of Mathematics, Isfahan (Khorasgan) Branch, Islamic Azad University, Isfahan P.O. Box 158-81595, Iran; zismeal@gmail.com (Z.I.S.); masoudallame@yahoo.com (M.A.)
[2] Information Technology Research and Development Center, University of Kufa, Najaf 540011, Iraq; mohammeds.abed@uokufa.edu.iq
[*] Correspondence: tavassoli_k@yahoo.com

**Abstract:** Proposing a matrix transform method to solve a fractional partial differential equation is the main aim of this paper. The main model can be transferred to a partial-integro differential equation (PIDE) with a weakly singular kernel. The spatial direction is approximated by a fourth-order difference scheme. Also, the temporal derivative is discretized via a second-order numerical procedure. First, the spatial derivatives are approximated by a fourth-order operator to compute the second-order derivatives. This process produces a system of differential equations related to the time variable. Then, the Crank–Nicolson idea is utilized to achieve a full-discrete scheme. The kernel of the integral term is discretized by using the Lagrange polynomials to overcome its singularity. Subsequently, we prove the convergence and stability of the new difference scheme by utilizing the Rayleigh–Ritz theorem. Finally, some numerical examples in one-dimensional and two-dimensional cases are presented to verify the theoretical results.

**Keywords:** matrix transform method; fourth-order difference scheme; partial-integro differential equation; Rayleigh–Ritz theorem; error estimate

**MSC:** 65M12; 65N12; 65N30; 65N40

## 1. Introduction

PIDEs, also known as partial-integro differential equations, are commonly used in the simulation of a wide range of complicated systems. In this work, we focused on PIDEs with a weakly singular kernel. These forms of equations are frequently encountered in applications such as heat flow in a material with memory [1] and linear viscoelastic mechanics [2,3]. Recently, several researchers developed a variety of techniques for numerically solving integro differential equations with a weakly singular kernel, for example, the Haar wavelet method [4], high-order ADI orthogonal spline collocation method [5], orthogonal cubic spline method [6], compact finite difference method [7], a fully spectral Galerkin method [8], space–time Muntz spectral collocation approach [9], differential-quadrature-based approach [10], two-grid temporal second-order scheme [11], second-order IMEX scheme [12,13], and ADI Galerkin finite element methods [14].

The finite difference method is the basis of the proposed matrix transform method, which may be coupled with other techniques. These approaches are used to solve a variety of PDEs and PIDEs, including a fourth-order PIDE with a weakly singular kernel [15], fractional PIDE of Volterra type [16], the PIDE that derives by financial stochastic processes [17], the PIDE obtained from the filtration model [18], and the time fractional PIDE [19]. Other approaches based on the matrix transform have been developed, such as Fourier–Bessel matrix transforms [20], Bernstein operational matrix [21], etc.

The most challenging part of PIDEs with a weakly singular kernel is dealing with the singular part of the kernel. The value of the kernel function may not be determined in some

grid points due to the singularity. One solution to this challenge involves selecting the base elements in such a way that eliminates the singular part of the kernel. Some of these bases can be mentioned here, such as hybrid block-pulse functions based on Legendre polynomials [22], multivariate Jacobi polynomials [23], shifted first-kind Chebyshev polynomials [24], and transformed fractional Jacobi polynomials [25]. To overcome singularity, we employ Lagrange interpolation for the integral part of equations and the Crank–Nicolson method for the temporal variable. The compact finite difference method (CFDM) is one of the improvements of the classic finite difference method (FDM) for solving linear and nonlinear PDEs, for example, two-dimensional Schrödinger–Boussinesq equations [26], time–space fractional differential equations [27], nonlinear Klein–Gordon equations [28], 2D anomalous sub-diffusion equations [29], etc.

## 2. Matrix Transform Technique

First, we consider the following definition.

**Definition 1.** *The left fractional integral of function $v \in H^1([a,b])$ with order $\alpha > 0$ is defined as*

$$_aD_t^{-\alpha}v(x) = \frac{1}{\Gamma(\alpha)} \int_a^t (t-y)^{\alpha-1}v(y)\mathrm{d}y. \tag{1}$$

Now, in the current paper, consider the following PDEs with fractional integral

$$\begin{cases} \dfrac{\partial u(\boldsymbol{x},t)}{\partial t} = \Delta u(\boldsymbol{x},t) +_0 D_t^{-\alpha}\Delta u(\boldsymbol{x},t) + f(\boldsymbol{x},s), & \boldsymbol{x} \in \Omega, \quad t \geqslant 0, \\ u(\boldsymbol{x},0) = g(x), \\ u(\boldsymbol{x},t) = p(t), & \boldsymbol{x} \in \partial\Omega. \end{cases} \tag{2}$$

The mathematical model Equation (2) can be rewritten to the following PIDE (it must be noted that we eliminate coefficient $\dfrac{1}{\Gamma(\alpha)}$)

$$\begin{cases} \dfrac{\partial u(\boldsymbol{x},t)}{\partial t} = \Delta u(\boldsymbol{x},t) + \displaystyle\int_0^t (t-s)^{\alpha-1}\Delta u(\boldsymbol{x},s)\,ds + f(\boldsymbol{x},s), & \boldsymbol{x} \in \Omega, \quad t \geqslant 0, \\ u(\boldsymbol{x},0) = g(x), \\ u(\boldsymbol{x},t) = p(t), & \boldsymbol{x} \in \partial\Omega, \end{cases} \tag{3}$$

where $u \in C^2(\Omega \times (0,T])$ and $0 < \alpha < 1$. Some applications of the fractional PDEs can be found in [30,31]. Also, Caputo [32] introduced the application of differential equations with fractional derivatives for generalizing stress–strain relations of unelastic media.

*Discrete Scheme of 2D Case*

To discretize the space derivatives, the following sets must be defined. Let

$$\begin{aligned} V_x &= \left\{ x_n = nh_x, \quad h_x = \frac{L}{M_x}, \qquad n = 0,1,2,\ldots,M_x \right\}, \\ V_y &= \left\{ y_m = mh_y, \quad h_y = \frac{L}{M_y}, \qquad m = 0,1,2,\ldots,M_y \right\}. \end{aligned}$$

where $h_x$ and $h_y$ are the step sizes of spatial variables. Also, we set $u_{n,m} = u(x_n, y_m, t)$. To approximate the spatial derivatives by the fourth-order operator, we define the following operators [33]

$$\delta_x^2 u_{n,m} = u_{n+1,m} - 2u_{n,m} + u_{n-1,m}, \tag{4}$$

$$\delta_y^2 u_{n,m} = u_{n,m+1} - 2u_{n,m} + u_{n,m-1}, \tag{5}$$

$$\mathcal{I}_x u_{n,m} = h_x^2 \left(1 + \frac{1}{12}\delta_x^2\right) u_{n,m}, \tag{6}$$

$$\mathcal{I}_y u_{n,m} = h_y^2 \left(1 + \frac{1}{12}\delta_y^2\right) u_{n,m}, \tag{7}$$

**Lemma 1** ([34]). *The fourth-order compact difference operators with maintaining three-point stencil to approximate the $u_{xx}(x_n, y_m, t)$ and $u_{yy}(x_n, y_m, t)$ are*

$$\frac{\delta_x^2}{h_x^2 \left(1 + \frac{1}{12}\delta_x^2\right)} u_{n,m} = \left.\frac{\partial^2 u}{\partial x^2}\right|_{(n,m)} - \frac{1}{240} \left.\frac{\partial^4 u}{\partial x^4}\right|_{(n,m)} h_x^4 + \mathcal{O}(h_x^6), \tag{8}$$

$$\frac{\delta_y^2}{h_y^2 \left(1 + \frac{1}{12}\delta_y^2\right)} u_{n,m} = \left.\frac{\partial^2 u}{\partial y^2}\right|_{(n,m)} - \frac{1}{240} \left.\frac{\partial^4 u}{\partial y^4}\right|_{(n,m)} h_x^4 + \mathcal{O}(h_y^6). \tag{9}$$

Applying relations Equations (8) and (9) in Equation (3) provides

$$\frac{dU_{n,m}}{dt} = \frac{\delta_x^2}{h_x^2 \left(1 + \frac{1}{12}\delta_x^2\right)} U_{n,m} + \frac{\delta_y^2}{h_y^2 \left(1 + \frac{1}{12}\delta_y^2\right)} U_{n,m} \tag{10}$$

$$+ \int_0^t (t-s)^{\alpha-1} \left(\frac{\delta_x^2}{h_x^2 \left(1 + \frac{1}{12}\delta_x^2\right)} U_{n,m} + \frac{\delta_y^2}{h_y^2 \left(1 + \frac{1}{12}\delta_y^2\right)} U_{n,m}\right) ds + f_{n,m}(t),$$

for $n = 1, \ldots, M_x - 1$, $m = 1, \ldots, M_y - 1$, where $U_{n,m} = u(x_n, y_m, t)$ and $f_{n,m} = f(x_n, y_m, t)$. The simplified form of Equation (10) is

$$\frac{d\mathcal{I}_x\mathcal{I}_y U_{n,m}}{dt} = \mathcal{I}_y \delta_x^2 u_{n,m} + \mathcal{I}_x \delta_y^2 u_{n,m} + \int_0^t (t-s)^{\alpha-1} \left(\mathcal{I}_y \delta_x^2 u_{n,m} + \mathcal{I}_x \delta_y^2 u_{n,m}\right) ds + \mathcal{I}_x\mathcal{I}_y f_{n,m}(t). \tag{11}$$

Now, we can derive the following matrix form

$$A\frac{d\boldsymbol{U}}{dt} = B\boldsymbol{U} + B \int_0^t (t-s)^{\alpha-1} \boldsymbol{U}(s) ds + A\boldsymbol{F}, \tag{12}$$

or

$$\frac{d\boldsymbol{U}}{dt} = A^{-1}B\boldsymbol{U} + A^{-1}B \int_0^t (t-s)^{\alpha-1} \boldsymbol{U}(s) ds + \boldsymbol{F}, \tag{13}$$

where

$$A = \left(h_x^2 h_y^2\right) \text{tridiag}(A_2, A_1, A_2), \tag{14}$$

$$A_1 = \text{tridiag}\left(\frac{5}{72}, \frac{25}{36}, \frac{5}{72}\right), \tag{15}$$

$$A_2 = \text{tridiag}\left(\frac{1}{144}, \frac{5}{72}, \frac{1}{144}\right), \tag{16}$$

$$B = \text{tridiag}(B_2, B_1, B_2), \tag{17}$$

$$B_1 = \text{tridiag}\left(\frac{5}{6}h_y^2 - \frac{1}{6}h_x^2, -\frac{10}{6}(h_x^2 + h_y^2), \frac{5}{6}h_y^2 - \frac{1}{6}h_x^2\right), \tag{18}$$

$$B_2 = \text{tridiag}\left(\frac{1}{12}h_x^2 + \frac{1}{12}h_y^2, \frac{5}{6}h_x^2 - \frac{1}{6}h_y^2, \frac{1}{12}h_x^2 + \frac{1}{12}h_y^2\right). \tag{19}$$

Matrices $A$, $B$ are block-Toeplitz tridiagonal (block-TT) matrices of order $(M_x - 1)(M_y - 1)$ and matrices $A_i$, $B_i$, $i = 1, 2$, are tridiagonal matrices of order $(M_x - 1)$. We can formulate each one as a sum of two Kronecker products of matrices [35]. For example, matrix $A$ can be written as

$$A = \left(h_x^2 h_y^2\right)\left[\left(I_{M_y-1} \otimes A_1\right) + \left(J_{M_y-1} \otimes A_2\right)\right], \tag{20}$$

where $I_{M_y-1}$ is the identity matrix and $J_{M_y-1} = \text{tridiag}(1, 0, 1)$. The eigenvalues of $A$ are obtained from the following relation [36]

$$\lambda_{kj} = \left(h_x^2 h_y^2\right)\left[\lambda_k^{(A_1)} + 2\lambda_k^{(A_2)} \cos\frac{j\pi}{M_y}\right], \quad j = 1, 2, \ldots, M_y - 1, \ k = 1, 2, \ldots, M_x - 1, \tag{21}$$

where $\{\lambda_k^{(A_1)}\}_{k=1}^{M_x-1}$, $\{\lambda_k^{(A_2)}\}_{k=1}^{M_x-1}$ are the eigenvalues of the $A_1$ and $A_2$, respectively, calculated using the following relations

$$\lambda_k^{(A_1)} = \frac{25}{36} + \frac{10}{72}\cos\left(\frac{k\pi}{M_x}\right), \tag{22}$$

$$\lambda_k^{(A_2)} = \frac{5}{72} + \frac{1}{72}\cos\left(\frac{k\pi}{M_x}\right), \quad k = 1, 2, \ldots, M_x - 1.$$

Note that the function $f(k) = \cos\left(\frac{k\pi}{M_x}\right)$ is a strictly descending on set $\{1, 2, \ldots M_x - 1\} := \Lambda$, so we have

$$\max_{k \in \Lambda} f(k) = f(1) = \cos\left(\frac{\pi}{M_x}\right), \quad \min_{k \in \Lambda} f(k) = f(M_x - 1) = \cos\left(\frac{(M_x - 1)\pi}{M_x}\right). \tag{23}$$

For the eigenvalues of $A$ matrix, let

$$\max_{k,j} \lambda_{kj} := \lambda_{max}^{(A)}, \quad \min_{k,j} \lambda_{kj} := \lambda_{min}^{(A)}.$$

Then, from Equations (21)–(23), we can conclude

$$\lambda_{max}^{(A)} = \left(h_x^2 h_y^2\right)\left[\left[\frac{25}{36}+\frac{10}{72}\cos\left(\frac{\pi}{M_x}\right)\right]+2\left[\frac{5}{72}+\frac{1}{72}\cos\left(\frac{\pi}{M_x}\right)\right]\cos\left(\frac{\pi}{M_y}\right)\right], \tag{24}$$

$$\lambda_{min}^{(A)} = \left(h_x^2 h_y^2\right)\left[\left[\frac{25}{36}+\frac{10}{72}\cos\left(\frac{(M_x-1)\pi}{M_x}\right)\right]+2\left[\frac{5}{72}+\frac{1}{72}\cos\left(\frac{(M_x-1)\pi}{M_x}\right)\right]\cos\left(\frac{(M_y-1)\pi}{M_y}\right)\right]. \tag{25}$$

The simplified forms of relations Equations (24) and (25) are

$$\lambda_{max}^{(A)} = \left(h_x^2 h_y^2\right)\left[\frac{1}{36}\left(5\cos\left(\frac{\pi}{M_x}\right)+25\right)+2\left(\frac{1}{72}\cos\left(\frac{\pi}{M_x}\right)+\frac{5}{72}\right)\cos\left(\frac{\pi}{M_y}\right)\right], \tag{26}$$

$$\lambda_{min}^{(A)} = \left(h_x^2 h_y^2\right)\left[\frac{1}{36}\left(5\cos\left(\frac{(M_x-1)\pi}{M_x}\right)+25\right)+2\left(\frac{1}{72}\cos\left(\frac{(M_x-1)\pi}{M_x}\right)+\frac{5}{72}\right)\cos\left(\frac{(M_y-1)\pi}{M_y}\right)\right]. \tag{27}$$

For the sake of simplicity, let $M_x = M_y := M$ and $h_x = h_y := h$; we can rewrite Equations (26) and (27) as

$$\lambda_{max}^{(A)} = \frac{h^4}{36}\left(25+10\cos\left(\frac{\pi}{M}\right)+\cos^2\left(\frac{\pi}{M}\right)\right), \tag{28}$$

$$\lambda_{min}^{(A)} = \frac{h^4}{36}\left(25+10\cos\left(\frac{(M-1)\pi}{M}\right)+\cos^2\left(\frac{(M-1)\pi}{M}\right)\right). \tag{29}$$

Similarly, we can indite matrix $B$ as

$$B = \left(I_{M_y}\otimes B_1\right)+\left(L_{M_y}\otimes B_2\right). \tag{30}$$

The eigenvalues of matrix $B$ are

$$\lambda_{kj} = \lambda_k^{(B_1)}+2\lambda_k^{(B_2)}\cos\frac{j\pi}{M_y}, \quad j=1,2,\ldots,M_y-1, \ \ k=1,2,\ldots,M_x-1, \tag{31}$$

where $\{\lambda_k^{(B_1)}\}_{k=1}^{M_x}$ and $\{\lambda_k^{(B_2)}\}_{k=1}^{M_x}$ are calculated using the following relations

$$\lambda_k^{(B_1)} = -\frac{10}{6}(h_x^2+h_y^2)+2\left(\frac{5}{6}h_y^2-\frac{1}{6}h_x^2\right)\cos\left(\frac{k\pi}{M_x}\right), \tag{32}$$

$$\lambda_k^{(B_2)} = \frac{5}{6}h_x^2-\frac{1}{6}h_y^2+2\left(\frac{1}{12}h_x^2+\frac{1}{12}h_y^2\right)\cos\left(\frac{k\pi}{M_x}\right), \quad k=1,2,\ldots,M_x-1. \tag{33}$$

To compute the eigenvalues of $B$ matrix, let

$$\max_{k,j}\ \lambda_{kj}:=\lambda_{max}^{(B)}, \qquad \min_{k,j}\ \lambda_{kj}:=\lambda_{min}^{(B)}.$$

Then, we have

$$\lambda_{max}^{(B)} = \left[-\frac{10}{6}(h_x^2 + h_y^2) + 2\left(\frac{5}{6}h_y^2 - \frac{1}{6}h_x^2\right)\cos\left(\frac{\pi}{M_x}\right)\right] \tag{34}$$

$$+ \quad 2\left[\frac{5}{6}h_x^2 - \frac{1}{6}h_y^2 + 2\left(\frac{1}{12}h_x^2 + \frac{1}{12}h_y^2\right)\cos\left(\frac{\pi}{M_x}\right)\right]\cos\left(\frac{\pi}{M_y}\right),$$

$$\lambda_{min}^{(B)} = \left[-\frac{10}{6}(h_x^2 + h_y^2) + 2\left(\frac{5}{6}h_y^2 - \frac{1}{6}h_x^2\right)\cos\left(\frac{(M_x - 1)\pi}{M_x}\right)\right] \tag{35}$$

$$+ \quad 2\left[\frac{5}{6}h_x^2 - \frac{1}{6}h_y^2 + 2\left(\frac{1}{12}h_x^2 + \frac{1}{12}h_y^2\right)\cos\left(\frac{(M_x - 1)\pi}{M_x}\right)\right]\cos\left(\frac{(M_y - 1)\pi}{M_y}\right).$$

The simplified forms of relations Equations (34) and (35) are

$$\lambda_{max}^{(B)} = \frac{1}{3}\left[-5\left(h_x^2 + h_y^2\right) + \left(5h_y^2 - h_x^2\right)\cos\left(\frac{\pi}{M_x}\right)\right. \tag{36}$$

$$+ \quad \left.\left(5h_x^2 + \left(h_x^2 + h_y^2\right)\cos\left(\frac{\pi}{M_x}\right) - h_y^2\right)\cos\left(\frac{\pi}{M_y}\right)\right],$$

$$\lambda_{min}^{(B)} = \frac{1}{3}\left[-5\left(h_x^2 + h_y^2\right) + \left(5h_y^2 - h_x^2\right)\cos\left(\frac{(M_x - 1)\pi}{M_x}\right)\right. \tag{37}$$

$$+ \quad \left.\left(5h_x^2 + \left(h_x^2 + h_y^2\right)\cos\left(\frac{(M_x - 1)\pi}{M_x}\right) - h_y^2\right)\cos\left(\frac{(M_y - 1)\pi}{M_y}\right)\right].$$

For the sake of simplicity, let $M_x = M_y := M$ and $h_x = h_y := h$; we can change Equations (36) and (37) as

$$\lambda_{max}^{(B)} = \frac{h^2}{3}\left(-10 + 8\cos\left(\frac{\pi}{M}\right) + 2\cos^2\left(\frac{\pi}{M}\right)\right), \tag{38}$$

$$\lambda_{min}^{(B)} = \frac{h^2}{3}\left(-10 + 8\cos\left(\frac{(M - 1)\pi}{M}\right) + 2\cos^2\left(\frac{(M - 1)\pi}{M}\right)\right). \tag{39}$$

## 3. Approximating an Integral Part with Lagrange Interpolation

First, we utilize the Crank–Nicolson method to discretize temporal variables in Equation (12). Let

$$t_n = n\tau, \quad \tau = \frac{T}{M}, \quad n = 0, 1, \ldots, M,$$

where $T$ is the final time. Equaiton (12) yields

$$\frac{U^n - U^{n-1}}{\tau} = \frac{A^{-1}BU^n + A^{-1}BU^{n-1}}{2} + A^{-1}B\int_0^{t_{n-\frac{1}{2}}}\left(t_{n-\frac{1}{2}} - s\right)^{\alpha-1}U(s)ds + F^{n-\frac{1}{2}}. \tag{40}$$

We used the Lagrange interpolation to eliminate the singular points of the integral kernel. This fact is proposed in [37]. First, consider the following integral

$$\mathbb{I}^\alpha f = \int_0^t (t - s)^{\alpha-1}f(s)\,ds, \tag{41}$$

where $0 < \alpha < 1$. Now, Equation (40) can be written as the following expression

$$\mathbb{I}^\alpha f\Big|_{t=t_n} = \sum_{k=0}^{n-1} \int_{t_k}^{t_{k+1}} (t_n - s)^{\alpha-1} f(s)\, ds. \tag{42}$$

We want to approximate $f(t)$ on all intervals, such as $[t_k, t_{k+1}]$, and thus define the notation

$$f^k(t) = f(t)\Big|_{[t_k, t_{k+1}]}.$$

Let $t_k = t_0^{(k)} < t_1^{(k)}, \ldots < t_{p-1}^{(k)}, t_p^{(k)} = t_{k+1}$; then, utilizing the Lagrange interpolation provides [37]

$$f^k(t) = \sum_{i=0}^{p} f\left(t_i^{(k)}\right) \ell_{k,i}(t), \tag{43}$$

where

$$\ell_{k,i}(t) = \prod_{\substack{j=0 \\ j \neq i}}^{p} \frac{t - t_j^{(k)}}{t_i^{(k)} - t_j^{(k)}}. \tag{44}$$

According to Equation (43), we have

$$
\begin{aligned}
\mathbb{I}^\alpha f|_{t=t_n} &= \sum_{k=0}^{n-1} \int_{t_k}^{t_{k+1}} (t_n - s)^{\alpha-1} \left( \sum_{i=0}^{p} f\left(t_i^{(k)}\right) \ell_{k,i}(s) \right) ds \\
&= \sum_{k=0}^{n-1} \int_{t_k}^{t_{k+1}} (t_n - s)^{\alpha-1} \left( \sum_{i=0}^{p} f\left(t_i^{(k)}\right) \prod_{\substack{j=0 \\ j \neq i}}^{p} \frac{s - t_j^{(k)}}{t_i^{(k)} - t_j^{(k)}} \right) ds = \sum_{k=0}^{n-1} \sum_{i=0}^{p} a_{i,n}^{(k)} f\left(t_i^{(k)}\right),
\end{aligned} \tag{45}
$$

in which [31]

$$a_{i,n}^{(k)} = \int_{t_k}^{t_{k+1}} (t_n - s)^{-\alpha} \ell_{k,i}(s)\, ds. \tag{46}$$

**Lemma 2.** *Let $f \in C^{p+1}([0, T])$ and $p \in \mathbb{N}$ and then we have*

$$\left| \mathbb{I}^\alpha f|_{t=t_n} - \mathbb{I}^\alpha f^k|_{t=t_n} \right| \leq C\tau^{p+1},$$

*where $C \in \mathbb{R}^+$.*

**Proof.** The use of interpolation error yields

$$\left| \mathbb{I}^\alpha f|_{t=t_n} - \mathbb{I}^\alpha f^k|_{t=t_n} \right| \leq \sum_{k=0}^{n-1} \int_{t_k}^{t_{k+1}} (t_n - s)^{\alpha-1} \left| f(t) - f^k(t) \right| ds$$

$$\leq \max_{t\in[0,t_n]} \left| f^{(p+1)}(t) \right| \frac{1}{(p+1)!} \sum_{k=0}^{n-1} \int_{t_k}^{t_{k+1}} (t_n-s)^{\alpha-1} \prod_{j=0}^{p} \left( s - t_j^{(k)} \right) ds \qquad (47)$$

$$\leq \max_{t\in[0,t_n]} \left| f^{(p+1)}(t) \right| \frac{(t_{k+1}-t_k)^{p+1}}{(p+1)!} \sum_{k=0}^{n-1} \int_{t_k}^{t_{k+1}} (t_n-s)^{\alpha-1} ds$$

$$= \max_{t\in[0,t_n]} \left| f^{(p+1)}(t) \right| \frac{t_n^\alpha}{(p+1)!(\alpha+1)} \tau^{p+1}.$$

$\square$

In this section, we approximate the integral part of Equation (40) by a second-order scheme

$$\int_0^{t_{n-\frac{1}{2}}} \left( t_{n-\frac{1}{2}} - s \right)^{\alpha-1} \mathbf{U}(s)\, ds = \frac{1}{2}\left[ \int_0^{t_n} (t_n-s)^{\alpha-1}\mathbf{U}(s)\, ds + \int_0^{t_{n-1}} (t_{n-1}-s)^{\alpha-1}\mathbf{U}(s)\, ds \right] + \mathcal{O}(\tau^2). \qquad (48)$$

In this moment, each integral part of the relation Equation (48) can be approximated by Equation (45), so

$$\int_0^{t_{n-\frac{1}{2}}} \left( t_{n-\frac{1}{2}} - s \right)^{\alpha-1} \mathbf{U}(s)\, ds = \frac{1}{2}\left[ \sum_{k=0}^{n-1}\sum_{i=0}^{p} a_{i,n}^{(k)} \mathbf{U}\left(t_i^{(k)}\right) + \sum_{k=0}^{n-2}\sum_{i=0}^{p} a_{i,n-1}^{(k)} \mathbf{U}\left(t_i^{(k)}\right) \right] + \mathcal{O}(\tau^2)$$

$$= \frac{1}{2}\left[ \sum_{k=0}^{n-1}\sum_{i=0}^{p} a_{i,n}^{(k)} \mathbf{U}^{k,i} + \sum_{k=0}^{n-2}\sum_{i=0}^{p} a_{i,n-1}^{(k)} \mathbf{U}^{k,i} \right] + \mathcal{O}(\tau^2), \qquad (49)$$

where $\mathbf{U}^{n-1,p} = \mathbf{U}^n$. Substituting relation Equation (49) into equality Equation (40) concludes

$$\frac{\mathbf{U}^n - \mathbf{U}^{n-1}}{\tau} = \frac{1}{2}\mathbf{R}\mathbf{U}^n + \frac{1}{2}\mathbf{R}\mathbf{U}^{n-1} + \frac{1}{2}\sum_{k=0}^{n-1}\sum_{i=0}^{p} a_{i,n}^{(k)}\mathbf{R}\mathbf{U}^{k,i} + \frac{1}{2}\sum_{k=0}^{n-2}\sum_{i=0}^{p} a_{i,n-1}^{(k)}\mathbf{R}\mathbf{U}^{k,i} + \mathbf{F}^{n-\frac{1}{2}}, \qquad (50)$$

where $\mathbf{R} = \mathbf{A}^{-1}\mathbf{B}$. In the end, we have

$$\mathbf{U}^n - \frac{\tau}{2}\left( I + a_{p,n}^{(n-1)}I \right)\mathbf{R}\mathbf{U}^n = \mathbf{U}^{n-1} + \frac{\tau}{2}\mathbf{R}\mathbf{U}^{n-1} + \frac{\tau}{2}a_{p,n-1}^{(k)}\mathbf{R}\mathbf{U}^{n-1}$$

$$+ \frac{\tau}{2}\sum_{k=0}^{n-2}\sum_{i=0}^{p} a_{i,n}^{(k)}\mathbf{R}\mathbf{U}^{k,i} + \frac{\tau}{2}\sum_{k=0}^{n-3}\sum_{i=0}^{p} a_{i,n-1}^{(k)}\mathbf{R}\mathbf{U}^{k,i} + \tau\mathbf{F}^{n-\frac{1}{2}}, \qquad (51)$$

and also

$$\left( I - \frac{\tau}{2}\left( 1 + a_{p,n}^{(n-1)} \right)\mathbf{R} \right)\mathbf{U}^n = \left( I + \frac{\tau}{2}\left( 1 + a_{p,n}^{(n-2)} + a_{p,n-1}^{(n-2)} \right)\mathbf{R} \right)\mathbf{U}^{n-1}$$

$$+ \frac{\tau}{2}\sum_{k=0}^{n-3}\sum_{i=0}^{p} a_{i,n}^{(k)}\mathbf{R}\mathbf{U}^{k,i} + \frac{\tau}{2}\sum_{k=0}^{n-3}\sum_{i=0}^{p} a_{i,n-1}^{(k)}\mathbf{R}\mathbf{U}^{k,i} + \tau\mathbf{F}^{n-\frac{1}{2}}. \qquad (52)$$

From the above relation, we can obtain

$$\mathbf{S}\mathbf{U}^n = \mathbf{Q}\mathbf{U}^{n-1} + \mathbf{G}^n, \qquad (53)$$

where

$$S \quad = \quad I - \frac{\tau}{2}\left(1 + a_{p,n}^{(n-1)}\right)R, \tag{54}$$

$$Q \quad = \quad I + \frac{\tau}{2}\left(1 + a_{p,n}^{(n-2)} + a_{p,n-1}^{(n-2)}\right)R, \tag{55}$$

$$G^n \quad = \quad \frac{\tau}{2}\sum_{k=0}^{n-3}\sum_{i=0}^{p} a_{i,n}^{(k)} R U^{k,i} + \frac{\tau}{2}\sum_{k=0}^{n-3}\sum_{i=0}^{p} a_{i,n-1}^{(k)} R U^{k,i} + \tau F^{n-\frac{1}{2}}. \tag{56}$$

## 4. Convergence and Stability Analysis

In order to discuss the convergence and stability of the proposed new method, the following preliminaries are necessary.

**Lemma 3** ([38]). *For every symmetric nonzero matrix $M$, the feature of the Rayleigh–Ritz (R–R) ratio is*

$$\lambda_{\min}(M) \le \frac{(M\Lambda, \Lambda)}{(\Lambda, \Lambda)} \le \lambda_{\max}(M),$$

*where $(X, Y)$ indicates the inner product and $\lambda_{min}$ and $\lambda_{max}$ denote smallest and largest eigenvalue, respectively.*

**Theorem 1.** *Let $U$ be smooth, sufficiently. The full-discrete scheme Equation (53) is unconditionally stable.*

**Proof.** Taking the inner product respect to $U^n$ results

$$S(U^n, U^n) = Q\left(U^{n-1}, U^n\right) + (G^n, U^n), \tag{57}$$

By utilizing the Cauchy–Schwartz inequality, we have

$$\begin{cases} S(U^n, U^n) \ge \lambda_{\min}(S)(U^n, U^n) = \lambda_{\min}(S)\|U^n\|^2, \\[2mm] \left|Q(U^{n-1}, U^n)\right| \le (QU^n, U^n)^{\frac{1}{2}}(QU^{n-1}, U^{n-1})^{\frac{1}{2}} \le \lambda_{\max}(Q)\|U^n\|\|U^{n-1}\|, \\[2mm] |(G^n, U^n)| \le \|G^n\|\|U^n\|. \end{cases} \tag{58}$$

The use of Equations (57) and (59) provides

$$\lambda_{\min}(S)\|U^n\|^2 \le \lambda_{\max}(Q)\|U^n\|\left\|U^{n-1}\right\| + \|G^n\|\|U^n\|.$$

Thus, we can obtain the following relation

$$\begin{aligned} \|U^n\| \quad &\le \quad \frac{\lambda_{\max}(Q)}{\lambda_{\min}(S)}\left\|U^{n-1}\right\| + \frac{1}{\lambda_{\min}(S)}\|G^n\| \\[3mm] &\le \quad \frac{\lambda_{\max}(Q)}{\lambda_{\min}(S)}\left\{\frac{\lambda_{\max}(Q)}{\lambda_{\min}(S)}\left\|U^{n-2}\right\| + \frac{1}{\lambda_{\min}(S)}\left\|G^{n-1}\right\|\right\} + \frac{1}{\lambda_{\min}(S)}\|G^n\| \\[3mm] &\vdots \\[3mm] &\le \quad \left(\frac{\lambda_{\max}(Q)}{\lambda_{\min}(S)}\right)^n\left\|U^0\right\| + \frac{1}{\lambda_{\min}(S)}\sum_{r=1}^{n}\left(\frac{\lambda_{\max}(Q)}{\lambda_{\min}(S)}\right)^{n-r}\|G^r\|. \end{aligned} \tag{59}$$

In the above relation

$$\lambda(S) = \left(1 - \frac{\tau}{2}\left(1 + a_{p,n}^{(n-1)}\right)\right)\lambda(R) = \left(1 - \frac{\tau}{2}\left(1 + a_{p,n}^{(n-1)}\right)\right)\frac{\lambda(B)}{\lambda(A)}, \tag{60}$$

$$\lambda(Q) = \left(1 + \frac{\tau}{2}\left(1 + a_{p,n}^{(n-2)} + a_{p,n-1}^{(n-2)}\right)\right)\lambda(R) = \left(1 + \frac{\tau}{2}\left(1 + a_{p,n}^{(n-2)} + a_{p,n-1}^{(n-2)}\right)\right)\frac{\lambda(B)}{\lambda(A)}. \tag{61}$$

Thus, we can conclude

$$\lambda_{\min}(S) = \left(1 - \frac{\tau}{2}\left(1 + a_{p,n}^{(n-1)}\right)\right)\lambda_{\min}(R) = \left(1 - \frac{\tau}{2}\left(1 + a_{p,n}^{(n-1)}\right)\right)\frac{\lambda_{\min}(B)}{\lambda_{\max}(A)}, \tag{62}$$

$$\lambda_{\max}(Q) = \left(1 + \frac{\tau}{2}\left(1 + a_{p,n}^{(n-2)} + a_{p,n-1}^{(n-2)}\right)\right)\lambda_{\max}(R) = \left(1 + \frac{\tau}{2}\left(1 + a_{p,n}^{(n-2)} + a_{p,n-1}^{(n-2)}\right)\right)\frac{\lambda_{\max}(B)}{\lambda_{\min}(A)}. \tag{63}$$

Corresponding to Equation (59), according to relations Equations (62) and (63) and using $\lim_{x\to\infty}\left(1 + \frac{1}{x}\right)^x = e$, we have

$$
\begin{aligned}
\|U^n\| &\leq \left(\frac{\lambda_{\max}(Q)}{\lambda_{\min}(S)}\right)^n\|U^0\| + \frac{1}{\lambda_{\min}(S)}\sum_{r=1}^{n}\left(\frac{\lambda_{\max}(Q)}{\lambda_{\min}(S)}\right)^{n-r}\|G^r\| \\
&\leq \left(\frac{\left(1 - \frac{\tau}{2}\left(1 + a_{p,n}^{(n-1)}\right)\right)\theta_1}{\left(1 + \frac{\tau}{2}\left(1 + a_{p,n}^{(n-2)} + a_{p,n-1}^{(n-2)}\right)\right)\theta_2}\right)^n\|U^0\| + \frac{1}{\lambda_{\min}(S)}\sum_{r=1}^{n}\left(\frac{\left(1 - \frac{\tau}{2}\left(1 + a_{p,n}^{(n-1)}\right)\right)\theta_1}{\left(1 + \frac{\tau}{2}\left(1 + a_{p,n}^{(n-2)} + a_{p,n-1}^{(n-2)}\right)\right)\theta_2}\right)^{n-r}\|G^r\| \\
&\leq \left(\frac{\exp\left(1 - \frac{T}{2}\left(1 + a_{p,n}^{(n-1)}\right)\right)\theta_1}{\exp\left(1 + \frac{T}{2}\left(1 + a_{p,n}^{(n-2)} + a_{p,n-1}^{(n-2)}\right)\right)\theta_2}\right)\|U^0\| + T\left(\frac{\exp\left(1 - \frac{T}{2}\left(1 + a_{p,n}^{(n-1)}\right)\right)\theta_1}{\exp\left(1 + \frac{T}{2}\left(1 + a_{p,n}^{(n-2)} + a_{p,n-1}^{(n-2)}\right)\right)\theta_2}\right)\max_{1\leq r\leq n}\|G^r\| \\
&\leq C_1\|U^0\| + C_2\max_{1\leq r\leq n}\|G^r\|,
\end{aligned}
\tag{64}
$$

which completes the proof; i.e., the developed scheme is unconditionally stable. □

**Theorem 2.** *Let $u^n$ and $U^n$ be exact and approximate solutions, respectively. The presented numerical scheme is convergent and the following inequality holds*

$$\|u^n - U^n\| \leq C\left(\tau^2 + h^4\right),$$

*where $C$ is a positive constant.*

**Proof.** The proof is similar to Theorem 1. □

**Remark 1.** *According to Atkinson's book, "Theoretical Numerical Analysis A Functional Analysis Framework", to obtain a robust numerical method for solving the PIDEs with weakly singular integral term based on the Lagrange interpolation, which preserves the order of accuracy, we must use a graded mesh approach. But, if we do not want to use this idea, we will have to make the time steps very small to obtain the order of theoretical convergence. Thus, in the current paper, we used the uniform meshes with very small time step in the numerical results. The use of uniform meshes isolates the singular points. It can be seen in the time-fractional PDEs, which are refined with the Graded Meshes Approach.*

## 5. Numerical Validations

*5.1. Example 1*

Here, the first considered test problem is

$$\begin{cases} \dfrac{\partial u(x,t)}{\partial t} = \Delta u(x,t) + \displaystyle\int_0^t (t-s)^{\alpha-1} \Delta u(x,s)\, ds + f(x,s), \quad x \in [x_l, x_r] = [0,1], \quad 0 < t \leq T, \\[3mm] u(x,0) = 0, \\[3mm] u(x_l, t) = 0, \qquad\qquad u(x_r, t) = 0, \end{cases} \tag{65}$$

where

$$f(x,t) = 2t\sin(\pi x) + \pi^2 \left( t^2 + \frac{2\Gamma(\alpha)}{\Gamma(3+\alpha)} t^{2+\alpha} \right) \sin(\pi x).$$

In this example, the exact solution is

$$u(x,t) = t^2 \sin(\pi x).$$

The current test problem is proposed to verify the theoretical topics. Table 1 demonstrates errors and computational rates in the spatial direction with $\tau = h^4$, $\alpha = 0.8$, $\alpha = 0.2$, and $T = 1$ for Example 1. Also, Table 2 shows errors and computational rates in the temporal direction with $\tau = h^2$, $\alpha = 0.1$, $\alpha = 0.9$, and $T = 1$ for Example 1. On the other hand, Tables 1 and 2 confirm the computational rates are near to the theoretical orders, i.e., second and fourth orders in the temporal and spatial directions, respectively.

**Table 1.** Errors and computational rates in spatial direction with $\tau = h^4$ for Example 1.

| | $\alpha = 0.8$ | | $\alpha = 0.2$ | |
|---|---|---|---|---|
| $h$ | $L_\infty$ | $C_2$-Order | $L_\infty$ | $C_2$-Order |
| 0.25 | $1.0078 \times 10^{-3}$ | – | $1.1779 \times 10^{-3}$ | – |
| 0.125 | $6.1334 \times 10^{-5}$ | 4.0383 | $6.9389 \times 10^{-5}$ | 4.0852 |
| 0.0625 | $3.8069 \times 10^{-6}$ | 4.0100 | $4.1919 \times 10^{-6}$ | 4.0490 |
| 0.03125 | $2.3751 \times 10^{-7}$ | 4.0026 | $2.5599 \times 10^{-7}$ | 4.0334 |
| 0.015625 | $1.4837 \times 10^{-8}$ | 4.0007 | $1.5724 \times 10^{-8}$ | 4.0251 |
| 0.0078125 | $9.1925 \times 10^{-10}$ | 4.0126 | $9.4757 \times 10^{-10}$ | 4.0526 |

**Table 2.** Errors and computational rates in temporal direction with $\tau = h^2$ for Example 1.

| | $\alpha = 0.1$ | | $\alpha = 0.9$ | |
|---|---|---|---|---|
| $h$ | $L_\infty$ | $C_2$-Order | $L_\infty$ | $C_2$-Order |
| 0.25 | $1.9945 \times 10^{-3}$ | – | $1.1779 \times 10^{-3}$ | – |
| 0.125 | $8.7132 \times 10^{-4}$ | 1.1947 | $6.9389 \times 10^{-3}$ | 1.7431 |
| 0.0625 | $2.5841 \times 10^{-4}$ | 1.7535 | $4.1919 \times 10^{-4}$ | 1.9424 |
| 0.03125 | $7.1229 \times 10^{-5}$ | 1.8591 | $2.5599 \times 10^{-4}$ | 1.9856 |
| 0.015625 | $1.9160 \times 10^{-5}$ | 1.8943 | $1.5724 \times 10^{-5}$ | 1.9962 |
| 0.0078125 | $5.0937 \times 10^{-6}$ | 1.9113 | $9.4757 \times 10^{-6}$ | 1.9989 |
| 0.0039062 | $1.3435 \times 10^{-6}$ | 1.9227 | $9.4757 \times 10^{-6}$ | 1.9997 |
| 0.0019531 | $3.5219 \times 10^{-7}$ | 1.9316 | $9.4757 \times 10^{-7}$ | 1.9999 |

*5.2. Example 2*

Now, for the 2D case, we investigate

$$\begin{cases} \dfrac{\partial u(x,y,t)}{\partial t} = \Delta u(x,y,t) + \displaystyle\int_0^t (t-s)^{\alpha-1} \Delta u(x,y,s)\, ds + f(x,y,s), & x \in \Omega, \quad 0 < t \le T, \\[4mm] u(x,y,0) = 0, \\[4mm] u(x,y,t) = t^\gamma e^{-\beta\left((x-x_0)^2 + (y-y_0)^2\right)}, & (x,y) \in \partial\Omega, \end{cases} \tag{66}$$

where

$$f(x,y,t) = \left\{ \gamma t^{\gamma-1} - \left( t^\gamma + t^{\gamma+\alpha} \frac{\Gamma(\alpha)\Gamma(1+\gamma)}{\Gamma(\alpha+\gamma+1)} \right) \left( \beta^2 (2x - 2x_0)^2 - 4\beta + \beta^2 (2y - 2y_0)^2 \right) \right\} e^{-\beta\left((x-x_0)^2 + (y-y_0)^2\right)},$$

then, the exact solution is

$$u(x,t) = t^\gamma e^{-\beta\left((x-x_0)^2 + (y-y_0)^2\right)}.$$

The used parameters for Tables 3 and 4 are $\Omega = [0,1] \times [0,1]$, $x_0 = y_0 = 0.5$, $\beta = -100$, and $\gamma = 2$. Table 3 reports errors and computational orders in the spatial direction with $\tau = h^4$, $\alpha = 0.1$, $\alpha = 0.5$, $\alpha = 0.8$, and $T = 1$ for Example 2. Table 2 displays errors and computational rates in the temporal direction with $\tau = h^2$, $\alpha = 0.15$, $\alpha = 0.65$, and $T = 1$ for Example 2. Consequently, Tables 3 and 4 affirm the computational orders are converging to the theoretical results, i.e., second and fourth orders in the temporal and spatial directions, respectively. Figure 1 depicts graph of approximate solution with $h = 1/400$, $\tau = 10^{-3}$, $T = 1$, and different values of $\beta$ for Example 2.

**Table 3.** Errors and computational rates in spatial direction for Example 2.

| | $h$ | $\tau$ | $L_2$ Error | $C_2$-Order | $L_\infty$ Error | $C_2$-Order |
|---|---|---|---|---|---|---|
| | $h = 0.25$ | $\tau = 0.25$ | $2.2924 \times 10^{-3}$ | – | $3.1207 \times 10^{-3}$ | – |
| $\alpha = 0.1$ | $h = 0.125$ | $\tau = 0.015625$ | $2.0381 \times 10^{-4}$ | 3.4916 | $2.7679 \times 10^{-4}$ | 3.4950 |
| | $h = 0.0625$ | $\tau = 0.0009765625$ | $1.2790 \times 10^{-5}$ | 3.9941 | $1.7486 \times 10^{-5}$ | 3.9845 |
| | $h = 0.03125$ | $\tau = 0.00006103515625$ | $6.9314 \times 10^{-7}$ | 4.2057 | $9.4800 \times 10^{-7}$ | 4.2052 |
| | $h = 0.25$ | $\tau = 0.25$ | $8.0112 \times 10^{-3}$ | – | $1.9290 \times 10^{-2}$ | – |
| $\alpha = 0.5$ | $h = 0.125$ | $\tau = 0.015625$ | $6.3321 \times 10^{-4}$ | 3.6613 | $8.6222 \times 10^{-4}$ | 3.6639 |
| | $h = 0.0625$ | $\tau = 0.0009765625$ | $4.1946 \times 10^{-5}$ | 3.9160 | $5.7428 \times 10^{-5}$ | 3.9082 |
| | $h = 0.03125$ | $\tau = 0.00006103515625$ | $2.6601 \times 10^{-6}$ | 3.9791 | $3.6460 \times 10^{-6}$ | 3.9774 |
| | $h = 0.25$ | $\tau = 0.25$ | $2.4333 \times 10^{-2}$ | – | $3.3311 \times 10^{-2}$ | – |
| $\alpha = 0.8$ | $h = 0.125$ | $\tau = 0.015625$ | $1.5717 \times 10^{-3}$ | 3.9525 | $2.1411 \times 10^{-3}$ | 3.9551 |
| | $h = 0.0625$ | $\tau = 0.0009765625$ | $9.8528 \times 10^{-5}$ | 3.9956 | $1.3492 \times 10^{-4}$ | 3.9883 |
| | $h = 0.03125$ | $\tau = 0.00006103515625$ | $6.1599 \times 10^{-6}$ | 3.9995 | $8.4462 \times 10^{-6}$ | 3.9976 |

**Table 4.** Errors and computational rates in temporal direction with $\tau = h^2$ for Example 2.

| | $\alpha = 0.65$ | | $\alpha = 0.15$ | |
|---|---|---|---|---|
| $h$ | $L_\infty$ | $C_2$-Order | $L_\infty$ | $C_2$-Order |
| 0.25 | $2.6407 \times 10^{-1}$ | – | $1.5778 \times 10^{-1}$ | – |
| 0.125 | $1.0110 \times 10^{-1}$ | 1.3851 | $4.3497 \times 10^{-2}$ | 1.8592 |
| 0.0625 | $2.6730 \times 10^{-2}$ | 1.9193 | $1.1088 \times 10^{-2}$ | 1.9720 |
| 0.03125 | $7.1119 \times 10^{-3}$ | 1.9101 | $2.7847 \times 10^{-3}$ | 1.9934 |
| 0.015625 | $1.7832 \times 10^{-3}$ | 1.9958 | $6.9696 \times 10^{-4}$ | 1.9984 |
| 0.0078125 | $4.4612 \times 10^{-4}$ | 1.9990 | $1.7429 \times 10^{-4}$ | 1.9996 |
| 0.0039062 | $1.1155 \times 10^{-4}$ | 1.9997 | $4.3576 \times 10^{-5}$ | 1.9999 |
| 0.0019531 | $2.7889 \times 10^{-5}$ | 1.9999 | $1.0894 \times 10^{-5}$ | 2.0000 |

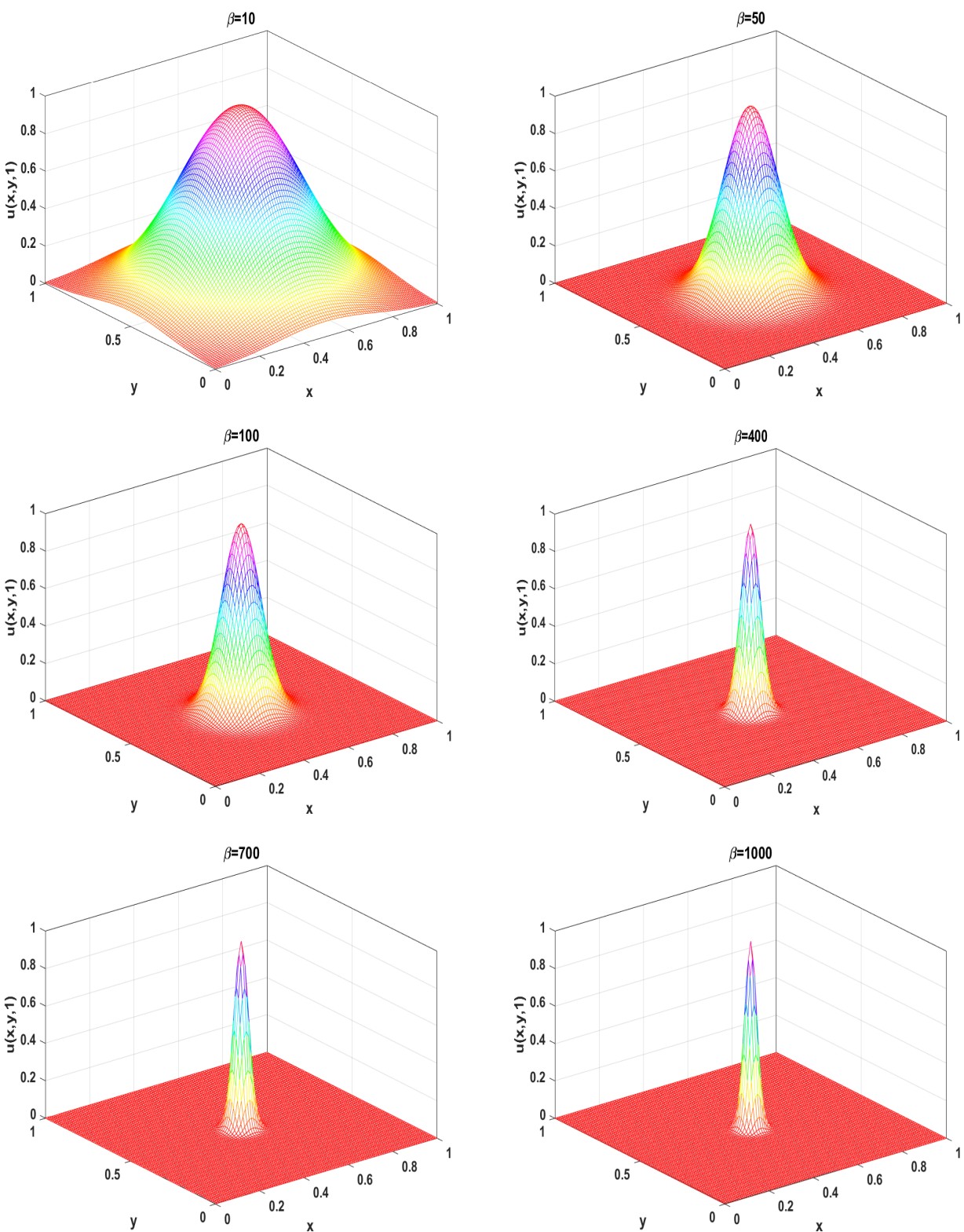

**Figure 1.** Graph of approximate solution with $h = 1/400$, $\tau = 10^{-3}$, and $T = 1$ for Example 2.

## 6. Conclusions

The current paper developed and analyzed a new matrix transform method to solve a PIDE with a singular kernel. First, the derivatives in $x$- and $y$-directions are approximated by utilizing a fourth-order operator and based upon a stencil with three points to obtain the full-discrete scheme. The mentioned procedure constructed a system of differential

equations related to the time variable. The eigenvalues of the coefficient matrices are derived explicitly. Also, the integral term is approximated by a Lagrange interpolation to overcome the singularity of the kernel. Then, a second-order difference approach is employed to obtain a full-discrete formulation. Since the eigenvalue of target matrices has been extracted, the Rayleigh–Ritz (R–R) ratio is engaged to prove the stability and convergence of the new numerical technique. Finally, two examples verify the capability and accuracy of the proposed numerical technique.

**Author Contributions:** Methodology, Z.I.S., M.T.K., M.S.M. and M.A.; Software, M.T.K.; Writing—review & editing, M.T.K. and M.A. All authors have read and agreed to the published version of the manuscript.

**Funding:** This research received no external funding.

**Data Availability Statement:** All the data is in the article.

**Conflicts of Interest:** The authors declare no conflict of interest.

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
