# Peer review of "Fourth-Order Difference Scheme and a Matrix Transform Approach for Solving Fractional PDEs"

_mathematics, doi:10.3390/math11173786_

Round 1

Author Response

The authors of the article thank the referee.

Reviewer 2 Report

This paper proposes a method for a partial-integro differential equation with weakly singular kernel. The temporal derivative is approximated by Crank-Nicolson technique to achieve 2nd order while the spatial derivative being approximated by well known compact finite difference to achieve 4th order. The convergence and stability are proved mainly via Ralyleigh-Ritz theorem. Numerical performance indeed supports established theoretical results.

1. There are many typos in this paper that need to be corrected, e.g. it looks that (2.1) may not be transferred into (2.2), expressions in (3.7), Lemma 4.1, and so on. I suggest authors check carefully and correct them all and other typos.

2. Some formulae are too wide, like (2.23)-(2.26), (4.5)-(4.8), Example 2, Table 3, and so on. I think it is better to make some changes.

3. The second and third inequalities in (4.8) are not obvious. More details are needed.

4. The authors mentioned that Lagrange polynomials can overcome singularity but there is no analysis and numerical illustrations about this part. More details are needed since it is very important in applications.

Check again.

Author Response

(The authors gave the same response as above.)

Round 2

Reviewer 2 Report

I think our questions are not well addressed.

1. I still find some inaccurate expressions, like from (2.1) to (2.2), the coefficient is not 1/(1-alpha). I think authors need to check and correct these misleading expressions.

2. I think authors may misunderstand [12]. Langrage interpolation in [12] is not on uniform grids. In general, Lagrange interpolation on uniform grids cannot overcome singularity. Therefore, more discussion is necessary.

I think our questions are not well addressed.

1. I still find some inaccurate expressions, like from (2.1) to (2.2), the coefficient is not 1/(1-alpha). I think authors need to check and correct these misleading expressions.

2. I think authors may misunderstand [12]. Langrage interpolation in [12] is not on uniform grids. In general, Lagrange interpolation on uniform grids cannot overcome singularity. Therefore, more discussion is necessary.

Author Response

(The authors gave the same response as above.)

Round 3

Reviewer 2 Report

I am fine with the authors' reply. 

I am fine with the English language.